# ARCHITECTURE MATTERS IN CONTINUAL LEARNING

## ABSTRACT

A large body of research in continual learning is devoted to overcoming the catastrophic forgetting of neural networks by designing new algorithms that are robust to the distribution shifts. However, the majority of these works are strictly focused on the *algorithmic* part of continual learning for a *fixed neural network architecture*, and the implications of using different architectures are not clearly understood. The few existing continual learning methods that expand the model also assume a fixed architecture and develop algorithms that can efficiently use the model throughout the learning experience. In contrast, in this work, we build on existing works that study continual learning from a neural network's architecture perspective and provide new insights into how the architecture choice, for the same learning algorithm, can impact stability-plasticity trade-off resulting in markedly different continual learning performance. We empirically analyze the impact of various architectural components providing best practices and recommendations that can improve the continual learning performance irrespective of the learning algorithm.

## 1 INTRODUCTION

Continual learning (CL) (Ring, 1995; Thrun, 1995) is a branch of machine learning where the model is exposed to a sequence of tasks with the hope of exploiting existing knowledge to adapt quickly to new tasks. The research in continual learning has seen a surge in the past few years with the explicit focus of developing algorithms that can alleviate *catastrophic forgetting* (McCloskey & Cohen, 1989)—whereby the model abruptly forgets the information of the past when trained on new tasks.

While most of the research in continual learning is focused on developing *learning algorithms*, that can perform better than naive fine-tuning on a stream of data, the role of model architecture, to the best of our knowledge, is not explicitly studied in any of the existing works. Even the class of parameter isolation or expansion-based methods, for example (Rusu et al., 2016; Yoon et al., 2018), only have a cursory focus on the model architecture insofar that they assume a specific architecture and develop an algorithm operating on the architecture. Orthogonal to this direction for designing algorithms, our motivation is that the inductive biases induced by different architectural components could be important for continual learning irrespective of the learning algorithm. Therefore, we seek to characterize the implications of different architectural choices in continual learning.

To motivate our study, consider a ResNet-18 model (He et al., 2016) on Split CIFAR-100, where CIFAR-100 dataset (Krizhevsky et al., 2009) is split into 20 disjoint sets—a prevalent architecture and benchmark in the existing continual learning works. Fig. 1a shows that explicitly designed CL algorithms, EWC (Kirkpatrick et al., 2017) (a parameter regularization-based method) and experience replay (Riemer et al., 2018) (a memory-based CL algorithm) indeed improve upon the naive fine-tuning. However, similar or better performance can be obtained on this benchmark by simply removing the global average pooling layer from ResNet-18 and performing the naive fine-tuning. This clearly demonstrates the need for a better understanding of network architectures in the context of continual learning where the architectural choices are not solely based on the performance of a single task but on a trade-off between the learning of new and previous tasks. Similar observation, though in more limited scenarios have been previously studied, for example Mirzadeh et al. (2022) looks at the role of layer width, while Ramasesh et al. (2022) focuses on the scale of the model. We build on these works, extending the analysis to architecture choices as well understanding particular components typically used like batch norm. It is also useful to note that these observations do not imply that the algorithmic improvements are not important. In fact, we show in Appendix B that one can achieve even better performance by combining our architectural findings with specially designed continual learning algorithms.

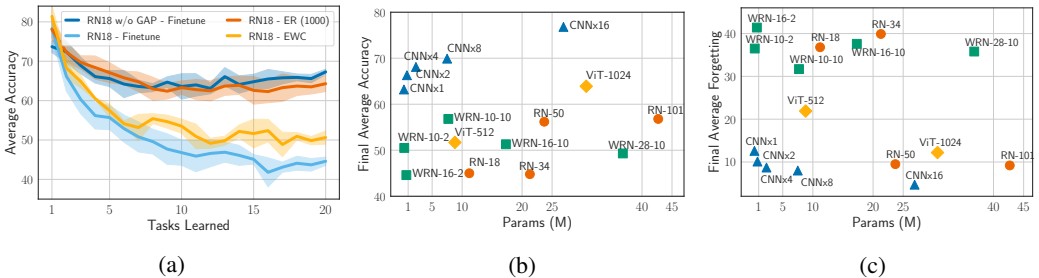

(a)                   (b)                   (c)

Figure 1: Split CIFAR-100: (a) While compared to naive fine-tuning, continual learning algorithms such as EWC and ER improve the performance, a simple modification to the architecture (removing global average pooling (GAP) layer) can match the performance of ER with a replay size of 1000 examples. (b) and (c) Different architectures lead to very different continual learning performance levels in terms of accuracy and forgetting. This work will investigate the reasons behind these gaps and provide insights into improving architectures.

To understand the implications of architectural decisions in continual learning, we thoroughly study different architectures including MLPs, CNNs, ResNets, Wide-ResNets and Visual Transformers. Our experiments suggest that different components of these architectures can have different effects on the relevant continual learning metrics—namely average accuracy, forgetting, and learning accuracy (*cf.* Sec. 2.1)—to the extent that vanilla fine-tuning with modified components can achieve similar of better performance than specifically designed CL methods on a given base architecture without significantly increasing the parameters count.

**Contributions.** We summarize our main contributions as follows:

- We compare both the learning and retention capabilities of popular architectures. We study the role of individual architectural decisions (e.g., width and depth, batch normalization, skip-connections, and pooling layers) and how they can impact the continual learning performance.
- We show that, in some cases, simply modifying the architecture can achieve a similar or better performance compared to specifically designed CL algorithms (on top of a base architecture).
- In addition to the standard CL benchmarks, Rotated MNIST and Split CIFAR-100, we report results on the large-scale Split ImageNet-1K benchmark, which is rarely used in the CL literature, to make sure our results hold in more complex settings.
- Inspired by our findings, we provide practical suggestions that are computationally cheap and can improve the performance of various architectures in continual learning.

**Limitations.** We emphasize that our main focus is to illustrate the significance of architectural decisions in continual learning. We do not claim that this work covers all the possible permutations of architectural components and different continual learning scenarios. Consequently, the majority of our experiments are focused on the task-incremental setup with popular architectures. However, our results in Appendix B.5 for the class-incremental setup confirm our results for the task-incremental setup. Moreover, the secondary aim of this work is to be a stepping-stone that encourages further research on the architectural side of continual learning. That is why we focus on the *breadth* rather than *depth* of some topics. Finally, while there are a limited number of works in the literature that study the role of architecture in continual learning, in Sec. 5 we will discuss why those works solely focus on specific topics while this work draws a comprehensive and general picture. We believe our work provides many interesting directions that require deeper analysis beyond the scope of this paper but can significantly improve our understanding of continual learning.

## 2    COMPARING ARCHITECTURES

### 2.1    EXPERIMENTAL SETUP

Here, for brevity, we explain our experimental setup but postpone more detailed information (e.g., hyper-parameters, details of architectures, etc.) to Appendix A.

**Benchmarks.** We use three continual learning benchmarks for our experiments. The Split CIFAR-100 includes 20 tasks where each task has the data of 5 classes (disjoint), and we train on each task for 10 epochs. The Split ImageNet-1K includes 10 tasks where each task includes 100 classes of

ImageNet-1K and we train on each task for 60 epochs. Finally, for a few experiments, we use the small Rotated MNIST benchmark with 5 tasks where the first task is the standard MNIST dataset, and each of the subsequent tasks adds 22.5 degrees of rotation to the images of the previous task. We note that the Split CIFAR-100 and Split ImageNet-1K benchmarks use a multi-head classification layer, while the MNIST benchmark uses a single-head classification layer. Thus, Split CIFAR-100 and Split ImageNet-1K belong to the so-called *task incremental learning* setting, whereas Rotated MNIST belongs to *domain incremental learning* (Hsu et al., 2018). Finally, for Split CIFAR-100 and Split ImageNet-1K benchmarks, we randomly shuffle the labels in each run, for 5 runs, to ensure that the results are not biased towards a specific dataset ordering.

**Architectures.** We denote each architecture with a descriptor. MLP-N represents fully connected networks with hidden layers of width N. Convolutions neural networks (CNN) are represented by CNN×N where N is the multiplier of the number of channels in each layer. Unless otherwise stated, the CNNs have only convolutional layers (with a stride of 2), followed by a dense feed-forward layer for classification. For the CIFAR-100 experiments, we use three convolutional layers, and for the ImageNet-1K experiments, we use six convolutional layers. Moreover, whenever we add pooling layers, we change the convolutional layer strides to 1 to keep the dimension of features the same. The standard ResNet (He et al., 2016) of depth D is denoted by ResNet-D and WideResNets (WRN) (Zagoruyko & Komodakis, 2016) are denoted by WRN-D-N where D and N are the depths and widths, respectively. Finally, we also use the recently proposed Vision Transformers (ViT) (Dosovitskiy et al., 2021). For the ImageNet-1K experiments, we follow the naming convention in the original paper (Dosovitskiy et al., 2021). However, for the Split CIFAR-100 experiments, we use smaller versions of ViTs where ViT N/M stands for a 4-layer vision transformer with the hidden size of N and MLP size of M. For *each architecture*, we search over a large grid of hyper-parameters and report the best results. Further, we average the results over 5 different random initializations, for the corresponding best hyper-parameters, and report the average and standard deviations.

**Metrics.** We are interested in comparing different architectures from two aspects: (1) how well an architecture can learn a new task *i.e.* their *learning ability* and (2) how well an architecture can preserve the previous knowledge *i.e.* their *retention ability*. For the former, we record *average accuracy*, *learning accuracy*, and *joint/ multi-task accuracy*, while for the latter we measure the *average forgetting* of the model. We now define these metrics.

(1) **Average Accuracy** $\in [0, 100]$ (the higher the better): The average validation accuracy after the model has been continually trained for $T$ tasks is defined as: $A_T = \frac{1}{T} \sum_{i=1}^{T} a_{T,i}$, where, $a_{t,i}$ is the validation accuracy on the dataset of task $i$ after the model finished learning task $t$.

(2) **Learning Accuracy** $\in [0, 100]$ (the higher the better): The accuracy for each task directly after it is learned. The learning accuracy provides a good representation of the plasticity of a model and can be calculated using: $\text{LA}_T = \frac{1}{T} \sum_{i=1}^{T} a_{i,i}$. Note that for both Split CIFAR-100 and ImageNet-1K benchmarks, since tasks include images with disjoint labels, the standard deviation of this metric can be high.

(3) **Joint Accuracy** $\in [0, 100]$ (the higher the better): The accuracy of the model when trained on the data of all tasks together.

(4) **Average Forgetting** $\in [-100, 100]$ (the lower the better): The average forgetting is calculated as the difference between the peak accuracy and the final accuracy of each task, after the continual learning experience is finished. For a continual learning benchmark with $T$ tasks, it is defined as: $F = \frac{1}{T-1} \sum_{i=1}^{T-1} \max_{t \in \{1,...,T-1\}} (a_{t,i} - a_{T,i})$.

## 2.2 RESULTS

We first compare different architectures on the Split CIFAR-100 and Split ImageNet-1K benchmarks. While this section broadly focuses on the learning and retention capabilities of different architectures, the explanations behind the performance gaps across different architectures and the analysis of different architectural components is given in the next section.

Tab. 1 lists the performance of different architectures on Split CIFAR-100 benchmark. One can make several observations from the table. First, very simple CNNs, which are not state-of-the-art (SOTA) in the single image classification tasks, significantly outperform both the ResNets, WRNs, and ViTs (all SOTA architectures in image classification) in terms of average accuracy and forgetting. This observation holds true for various sizes of widths and depths in all the architectures. A similar overall trend where, for a given parameter count, simple CNNs outperform other architectures, can also be seen in Fig. 1b and 1c. This shows that architectures that are cross-validated for a single task in an IID fashion are not necessarily optimal for continual learning settings.

Table 1: Split CIFAR-100: the learning and retention capabilities can vary significantly across different architectures.

Table 2: Split ImageNet-1K: the learning and retention capabilities can vary significantly across different architectures.

| Model | Params (M) | Average Accuracy | Average Forgetting | Learning Accuracy |
|---|---|---|---|---|
| CNN x1 | 0.3 | 62.2 ±1.35 | 12.6 ±1.14 | 74.1 ±7.72 |
| CNN x2 | 0.8 | 66.3 ±1.12 | 10.1 ±0.98 | 75.8 ±7.2 |
| CNN x4 | 2.3 | 68.1 ±0.5 | 8.7 ±0.21 | 76.4 ±6.92 |
| CNN x8 | 7.5 | 69.9 ±0.62 | 8.0 ±0.71 | 77.5 ±6.78 |
| CNN x16 | 26.9 | 76.8 ±0.76 | 4.7 ±0.84 | 81.0 ±6.97 |
| ResNet-18 | 11.2 | 45.0 ±0.63 | 36.8 ±1.08 | 74.9 ±3.98 |
| ResNet-34 | 21.3 | 44.8 ±2.34 | 39.9 ±2.28 | 72.6 ±6.34 |
| ResNet-50 | 23.6 | 56.2 ±0.88 | 9.5 ±0.38 | 67.8 ±5.09 |
| ResNet-101 | 42.6 | 56.8 ±1.62 | 9.2 ±0.89 | 65.7 ±5.42 |
| WRN-10-2 | 0.3 | 50.5 ±2.65 | 36.5 ±2.74 | 84.5 ±5.04 |
| WRN-10-10 | 7.7 | 56.8 ±2.03 | 31.7 ±1.34 | 86.7 ±4.94 |
| WRN-16-2 | 0.7 | 44.6 ±2.81 | 41.4 ±1.43 | 82.4 ±6.09 |
| WRN-16-10 | 17.3 | 51.3 ±1.47 | 37.6 ±2.22 | 86.9 ±3.96 |
| WRN-28-2 | 5.9 | 46.6 ±2.27 | 39.5 ±2.29 | 82.5 ±6.26 |
| WRN-28-10 | 36.7 | 49.3 ±2.02 | 35.8 ±2.56 | 82.5 ±6.26 |
| ViT-512/1024 | 8.8 | 51.7 ±1.4 | 21.9 ±1.3 | 71.4 ±5.52 |
| ViT-1024/1546 | 30.7 | 60.4 ±1.56 | 12.2 ±1.12 | 67.4 ±5.57 |

| Model | Params (M) | Average Accuracy | Average Forgetting | Learning Accuracy |
|---|---|---|---|---|
| CNN x3 | 9.1 | 63.3 ±0.68 | 5.4 ±0.93 | 71.6 ±2.31 |
| CNN x6 | 24.2 | 66.7 ±0.62 | 3.9 ±0.86 | 70.1 ±3.21 |
| CNN x12 | 72.4 | 67.8 ±1.04 | 2.8 ±0.7 | 70.3 ±2.82 |
| ResNet-34 | 21.8 | 62.7 ±0.53 | 17.3 ±0.58 | 78.4 ±2.57 |
| ResNet-50 | 25.5 | 66.1 ±0.69 | 19.0 ±0.67 | 83.3 ±1.57 |
| ResNet-101 | 44.5 | 64.1 ±0.72 | 18.9 ±1.32 | 81.1 ±2.89 |
| WRN-50-2 | 68.9 | 63.2 ±1.61 | 21.7 ±1.73 | 85.8 ±1.65 |
| ViT-Base | 86.1 | 58.3 ±0.65 | 15.9 ±1.11 | 72.8 ±2.25 |
| ViT-Large | 307.4 | 60.7 ±1.31 | 10.6 ±1.1 | 73.2 ±2.12 |

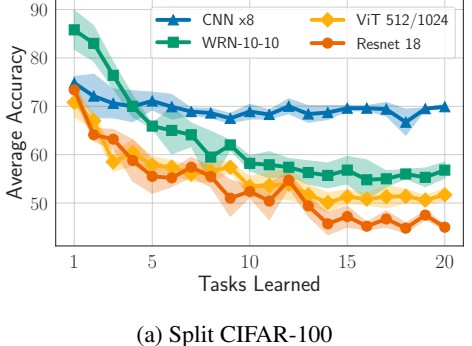

(a) Split CIFAR-100

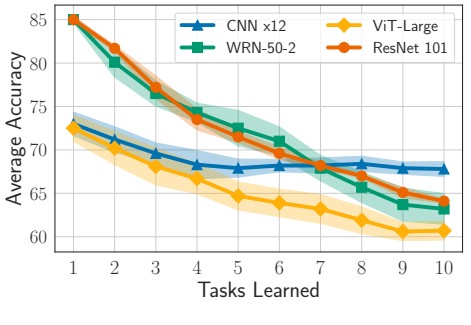

(b) Split ImageNet-1K

Figure 2: Evolution of average accuracy for various architectures on (a) Split CIFAR-100: CNNs have smaller forgetting than other architectures while WideResNets have the highest learning accuracy, and (b) Split ImageNet-1K WideResNets and ResNets have higher learning accuracy than CNNs and ViTs. However, the latter has smaller forgetting.

Second, a mere increase in the parameters count, within or across the architectures, does not necessarily translate into the performance increase in continual learning. For instance, ResNet-18 and ResNet-34 have roughly the same performance despite almost twice the number of parameters in the latter. Similarly, WRN-10-10 outperforms WRN-16-10 and WRN 28-10, despite having significantly less number of parameters. Note that we do not draw a general principle that overparametrization is not helpful in continual learning. In fact, in some cases, it indeed is helpful as can be in the across-the-board performance improvement when ResNet-34 is compared with ResNet-50 or when the WRN-10-2 is compared to the WRN-10-10. In the next section, we will analyze when the overparametrization can help the performance in continual learning.

Finally, explicitly comparing the learning and retention capabilities of different architectures, one can see from the table that ResNets and WRNs have a higher learning accuracy suggesting that they are better at learning a new task. This also explains their frequent use in single task settings. However, in terms of retention, CNNs and ViTs are much better, as evidenced by their lower forgetting numbers. This is further demonstrated in Fig. 2a (CIFAR-100) and Fig. 2b (ImageNet-1K), where ResNets and WRNs learn each individual task much better resulting in a higher average accuracy for the first few tasks. However, as the number of tasks increases, CNNs outperforms the other architectures due to their smaller forgetting, eventually translating into an overall flatter average accuracy curve.

A trend similar to CIFAR-100 can also be seen in the ImageNet-1K benchmark as shown in Table 2. However, the performance difference, as measured by the average accuracy, between CNNs and other architectures is smaller compared to that of CIFAR-100. We believe that this is due to the very high learning accuracy of other architectures compared to CNNs on this benchmark, and hence their

Table 3: Role of width and depth: increasing the number of parameters (by increasing width) reduces the forgetting and hence increases the average accuracy. However, increasing the depth does not necessarily improve the performance, and thus, it is essential to distinguish between scaling the models by making them deeper and wider.

| Benchmark | Model | Depth | Params (M) | Average Accuracy | Average Forgetting | Learning Accuracy |
|---|---|---|---|---|---|---|
| Rot MNIST | MLP-128 | 2 | 0.1 | 70.8 ±0.68 | 31.5 ±0.92 | 96.0 ±0.90 |
| Rot MNIST | MLP-128 | 8 | 0.2 | 68.9 ±1.07 | 35.4 ±1.34 | 97.3 ±0.76 |
| Rot MNIST | MLP-256 | 2 | 0.3 | 71.1 ±0.43 | 31.4 ±0.48 | 96.1 ±0.82 |
| Rot MNIST | MLP-256 | 8 | 0.7 | 70.4 ±0.61 | 32.1 ±0.75 | 96.3 ±0.77 |
| Rot MNIST | MLP-512 | 2 | 0.7 | 72.6 ±0.27 | 29.6 ±0.36 | 96.4 ±0.73 |
| CIFAR-100 | CNN x4 | 3 | 2.3 | 68.1 ±0.5 | 8.7 ±0.21 | 76.4 ±6.92 |
| CIFAR-100 | CNN x4 | 6 | 5.4 | 62.9 ±0.86 | 12.4 ±1.62 | 77.7 ±5.49 |
| CIFAR-100 | CNN x8 | 3 | 7.5 | 69.9 ±0.62 | 8.0 ±0.71 | 77.5 ±6.78 |
| CIFAR-100 | CNN x8 | 6 | 19.9 | 66.5 ± 1.01 | 10.7 ±1.19 | 76.6 ±4.78 |
| CIFAR-100 | ViT 512/1024 | 2 | 4.6 | 56.4 ±1.14 | 15.9 ±0.95 | 68.1 ±7.15 |
| CIFAR-100 | ViT 512/1024 | 4 | 8.8 | 51.7 ±1.4 | 21.9 ±1.3 | 71.4 ±5.52 |

frequent use in the single task settings, resulting in an improved final average accuracy. The average forgetting of CNNs is still much smaller than other architectures.

Overall, from both tables, we conclude that ResNets and WRNs have better learning abilities, whereas CNNs and ViTs have better retention abilities. In our experiments, simple CNNs achieve the best trade-off between learning and retention. In the next section, we study the individual components of these architectures to better understand their performance gap with respect to each other.

## 3   ROLE OF ARCHITECTURE COMPONENTS

We now study the individual components in various architectures to understand how they impact continual learning performance. We start by generic *structural* properties in all architectures such as *width* and *depth* (*cf.* Sec. 3.1), and show that as the width increases, the forgetting decreases. In Sec. 3.2, we study the impact of batch normalization and observe that it can significantly improve the learning accuracy in continual learning. Then, in Sec. 3.3, we see that adding skip connections (or shortcuts) to CNNs does not necessarily improve the CL performance whereas pooling layers (*cf.* Sec. 3.4 and  Sec. 3.5) can have significant impact on learning accuracy and forgetting. Moreover, we briefly study the impact of attention heads in ViTs in Sec. 3.6. Finally, based on the observations we make in the aforementioned sections, in Sec. 4, we provide a summary of modifications that can improve various architectures on both Split CIFAR-100 and ImageNet-1K benchmarks[1].

### 3.1   WIDTH AND DEPTH

Tab. 3 shows that across all architectures, over-parametrization through increasing width is helpful in improving the continual learning performance as evidenced by lower forgetting and higher average accuracy numbers. For MLP, when the width is increased from 128 to 512, the performance in all metrics improves. However, for both MLP-128 and MLP-256 when the depth is increased from 2 to 8 the average accuracy is reduced, and the average forgetting is increased with a marginal gain in learning accuracy. Finally, note that MLP-256 with 8 layers has roughly the same number of parameters as the MLP-512 with 2 layers. However, the wider MLP-512 has a better continual learning performance.

A similar analysis for ResNets and WideResNets is demonstrated in Tab. 1. ResNet-50 and ResNet-101 are four times wider than ResNet-18 and ResNet-34, and from the table, it can be seen that this width translates into drastic improvements in average accuracy and forgetting. Similarly, ResNet-34 and ResNet-101 are the deeper versions of ResNet-18 and ResNet-50, respectively. We can observe that increasing the depth is not helpful in this case either. Finally, wider WRN-10-10, WRN-16-10, and WRN-28-10 outperform the narrower WRN-10-2, WRN-10-10, WRN-28-10, respectively. Whereas if we fix the width, increasing the depth is not helpful. Overall, we can see that overparametrization through width is helpful in continual learning, whereas a similar claim cannot be made for the depth.

We note that Mirzadeh et al. (2022) studied the same width vs. depth trade-off in the task-incremental setting and demonstrated that overparametrization through width induces lazy-training regime during training and makes the gradients of different tasks more orthogonal and sparser, ultimately resulting in

---

[1]In this section, we duplicate some of the results across tables to improve readability.

Table 4: Role of various components for the Split CIFAR-100 benchmark: While adding skip connections does not have a significant impact on the performance, batch normalization and max polling can significantly improve the learning accuracy of CNNs.

| Model | Params (M) | Average Accuracy | Average Forgetting | Learning Accuracy | Joint Accuracy |
|---|---|---|---|---|---|
| CNN x4 | 2.3 | 68.1 ±0.5 | 8.7 ±0.21 | 76.4 ±6.92 | 73.4 ±0.89 |
| CNN x4 + Skip | 2.4 | 68.2 ±0.56 | 8.9 ±0.72 | 76.6 ±7.07 | 73.8 ±0.47 |
| CNN x4 + BN | 2.3 | 74.0 ±0.56 | 8.1 ±0.35 | 81.7 ±6.68 | 80.2 ±0.16 |
| CNN x4 + AvgPool | 2.3 | 68.5 ±0.6 | 8.3 ±0.57 | 76.3 ±7.63 | 73.6 ±0.83 |
| CNN x4 + MaxPool | 2.3 | 74.4 ±0.34 | 9.3 ±0.47 | 83.3 ±6.1 | 79.9 ±0.53 |
| CNN x4 + All | 2.4 | 77.7 ±0.77 | 6.5 ±0.58 | 83.7 ±6.31 | 81.6 ±0.77 |
| CNN x8 | 7.5 | 69.9 ±0.62 | 8.0 ±0.71 | 77.5 ±6.78 | 74.1 ±0.83 |
| CNN x8 + Skip | 7.8 | 70.7 ±0.31 | 6.8 ±0.91 | 77.1 ±6.87 | 74.4 ±0.35 |
| CNN x8 + BN | 7.5 | 76.1 ±0.3 | 5.9 ±0.16 | 81.7 ±6.83 | 80.5 ±0.27 |
| CNN x8 + AvgPool | 7.5 | 71.2 ±0.5 | 8.3 ±0.35 | 79.0 ±7.05 | 74.0 ±1.02 |
| CNN x8 + MaxPool | 7.5 | 77.2 ±0.53 | 7.1 ±0.33 | 84.0 ±5.81 | 80.6 ±0.35 |
| CNN x8 + All | 7.8 | 78.1 ±1.15 | 5.7 ±0.36 | 83.3 ±6.27 | 81.9 ±0.51 |
| CNN x16 | 26.9 | 76.8 ±0.76 | 4.7 ±0.84 | 81.0 ±6.97 | 79.1 ±0.86 |
| CNN x16 + All | 27.9 | 78.9 ±0.27 | 4.5 ±0.36 | 82.9 ±6.48 | 82.1 ±0.46 |

better continual learning performance. Our results verify their conclusions on more benchmarks and architectures. Moreover, we report the benefits of width in class-incremental setup in Appendix B.5.

**Conclusion.** Overall, in continual learning setups, over-parameterization thorough width can increase the accuracy and decrease the forgetting across all architectures. Given the same computing budget, wider and shallower networks can outperform thinner and deeper networks.

## 3.2 BATCH NORMALIZATION

Batch Normalization (BN) (Ioffe & Szegedy, 2015) is a normalization scheme that is shown to increase the convergence speed of the network due to its optimization and generalization benefits (Santurkar et al., 2018; Bjorck et al., 2018). Another advantage of the BN layer is its ability to reduce the covariate shift problem that is specifically relevant for continual learning where the data distribution may change from one task to the next. There are relatively few works that have studied the BN in the context of continual learning. Mirzadeh et al. (2020) analyzed the BN in continual learning through the generalization lens. Concurrently to this work, Pham et al. (2022) study the normalization schemes in continual learning and show that BN enables improved learning of each task. Additionally, the authors showed that in the presence of a replay buffer of previous tasks, BN facilitates a better knowledge transfer compared to other normalization schemes such as Group Normalization (Wu & He, 2018).

Intuitively, however, one might think that since due to evolving data distribution the BN statistics are changing across tasks and the statistics of each task are not kept, the BN should contribute to an increased forgetting. This is not the case in some of the experiments that we conducted. Similar to the results in Mirzadeh et al. (2020); Pham et al. (2022), we found the BN to facilitate the learning accuracy in Split CIFAR-100 and split ImageNet-1K (*cf.* Tab. 4 and Tab. 7).

We believe that this could be due to relatively unchanging BN statistics across tasks in these datasets. To verify this, in Appendix B.2, we plot the BN statistics of the first layer of CNN×4 on the Split CIFAR-100 dataset, and we show that the BN statistics are stable throughout the continual learning experience. However, if this hypothesis were to be true, the converse – a benchmark where the BN statistics change a lot across tasks, such as Permuted MNIST – should hurt the continual learning performance. In Appendix B.3, we plot the BN statistics of the first layer of MLP-128 on Permuted MNIST. It can be seen from the figure that indeed the BN statistics are changing in this benchmark. As a consequence, adding BN to this benchmark significantly hurt the performance, as evidenced by the increased forgetting in Tab. 9.

**Conclusion.** The effect of the batchnorm layer is data-dependent. In the setups where the input distribution relatively stays stable, such as Split CIFAR-100 or Split ImageNet-1K, the BN layer improves the continual learning performance by increasing the learning capability of the models. However, for setups where the input distribution changes significantly across tasks, such as Permuted MNIST, the BN layer can hurt the performance by increasing the forgetting.

### 3.3 Skip Connections

Skip connections (Cho et al., 2011), originally proposed for convolutional models by He et al. (2016), are crucial in the widely used ResNet architecture. They are also used in many other architectures such as transformers (Vaswani et al., 2017). Many works have been done to explain why skip connections are useful: Hardt & Ma (2016) show that skip connection tends to eliminate spurious local optima; Bartlett et al. (2018b) study the function expressivity of residual architectures; Bartlett et al. (2018a) show that gradient descent provably learns linear transforms in ResNets; Jastrzebski et al. (2017) show that skip connections tend to iteratively refine the learned representations.

However, these works mainly focus on learning a single task. In continual learning problems, due to the presence of distribution shift, it is unclear whether these benefits of skip connections still have a significant impact on model performance, such as forgetting and average accuracy over tasks. We empirically study the impact of skip connection on continual learning problems. Interestingly, as illustrated in Tab. 4, adding skip connections to plain CNNs does not change the performance significantly, and the results are very close (within the standard deviation) of each other.

**Conclusion.** The skip connection does not show a significant positive or negative impact on the model performance in our benchmarks.

### 3.4 Pooling Layers

Pooling layers were the mainstay of the improved performance of CNNs before ResNets. Pooling layers not only add local translation invariances, which help in applications like object classification (Krizhevsky et al., 2017; Dai et al., 2021), but also reduce the spatial resolution of the network features, resulting in the reduction of computational cost. Since one family of the architectures that we study are all-convolutional CNNs, we revisit the role of pooling layers in these architectures in a continual learning setup.

We compare the network without pooling, CNN×N , against those that have pooling layers ('CNN×N +AvgPool' or 'CNN×N +MaxPool') in Tab. 4. To keep the feature dimensions fixed, we set the convolutional stride from 2 to 1 when pooling is used. We make the following observations from the table. First, the average pooling (+AvgPool) does not have any significant impact on the continual learning metrics. Second, max pooling (+MaxPool) improves the learning capability of the network significantly, as measured by the improved learning accuracy. Third, in terms of retention, pooling layers do not have a significant impact as measured by similar forgetting. All in all, max pooling achieves the best average accuracy, owing to its superior learning capability.

The ability of max pooling to achieve better performance in a continual learning setting can be attributed to a well-known empirical observation by Springenberg et al. (2015), where it is shown that max pooling with stride 1 outperforms a CNN with stride 2 and no pooling. Further, we believe that max pooling might have extracted the low-level features, such as edges, better, resulting in improved learning in a dataset like CIFAR-100 that consists of natural images. There is some evidence in the literature that max pooling provides sparser features and precise localization (Zhou et al., 2016). This could have transferred over to the continual learning setup that we considered.

**Conclusion.** We believe the ability of max pooling for extracting low-level features, makes them a suitable choice for increasing the learning capability of models, especially on datasets consisting of natural images. Moreover, the intrinsic sparsity induced by max pooling in network may be helpful in continual learning scenarios. However, it remains an interesting future direction to further study the gains brought by max pooling in both the standard and continual learning setups.

### 3.5 Global Pooling Layers

Global average pooling (GAP) layers are typically used in convolutional networks just before the final classification layer to reduce the number of parameters in the classifier. The consequence of adding a GAP layer is to reduce the width of the final classifier. It is argued in Sec. 3.1 that wider networks forget less. Consequently, the architectures with a GAP layer can suffer from increased forgetting. Tab. 5 empirically demonstrate this intuition. From the table it can be seen that applying the GAP layer significantly increases the forgetting resulting in a lower average accuracy. In the previous section, we already demonstrated that average pooling does not result in a performance decrease as long as the spatial feature dimensions are the same. To demonstrate that there is nothing inherent to the GAP layer, and it is just a consequence of a reduced width of the final classifier, we construct another baseline by multiplying the number of channels in the last convolutional layer by 16 and then apply the GAP. This network is denoted as "CNN x4 (16x)" in Tab. 5 and it has the same

Table 5: Role of Global Average Pooling (GAP) for Split CIFAR-100: related to our arguments in Sec. 3.1, adding GAP to CNNs significantly increases the forgetting. Later, we show that removing GAP from ResNets can also significantly reduce forgetting as well.

| Model | Params (M) | Pre-Classification Width | Average Accuracy | Average Forgetting | Learning Accuracy | Joint Accuracy |
|---|---|---|---|---|---|---|
| CNN x4 | 2.3 | **8192** | 68.1 ±0.5 | 8.7 ±0.21 | 76.4 ±6.92 | 73.4 ±0.89 |
| CNN x4 + GAP | 1.5 | **512** | 60.1 ±0.43 | 14.3 ±0.8 | 66.1 ±7.76 | 76.9 ±0.81 |
| CNN x4 (16x) + GAP | 32.3 | **8192** | 73.6 ±0.39 | 5.2 ±0.66 | 75.6 ±4.77 | 77.9 ±0.37 |
| CNN x8 | 7.5 | **16384** | 69.9 ±0.62 | 8.0 ±0.71 | 77.5 ±6.78 | 74.1 ±0.83 |
| CNN x8 + GAP | 6.1 | **1024** | 63.1 ±2.0 | 14.7 ±1.68 | 70.1 ±7.18 | 78.3 ±0.97 |
| CNN x16 | 26.9 | **32768** | 76.8 ±0.76 | 4.7 ±0.84 | 81.0 ±6.97 | 74.6 ±0.86 |
| CNN x16 + GAP | 23.8 | **2048** | 66.3 ±0.82 | 12.2 ±0.65 | 72.3 ±6.02 | 78.9 ±0.27 |

classifier width as that of the network without GAP. It can be seen this architecture has considerably smaller forgetting showing GAP affects continual learning through the width of the final classifier.

**Conclusion.** Global pooling layers, while being computationally beneficial, reduce the dimensionality of the final features resulting in low continual learning performance. A practical workaround could be to use pooling layers with smaller filter sizes. Inspired by this observation, in Sec. 4 we show that the performance of ResNets in continual learning can be significantly improved by either removing the GAP layer or using the smaller average pooling window rather than the GAP.

### 3.6 ATTENTION HEADS

The attention mechanism is a prominent component in the transformer-based architectures Vaswani et al. (2017) that have had great success in natural language processing and, more recently, computer vision tasks. For the latter, the heads of vision transformers (ViTs) are shown to attend to both local and global features in the image Dosovitskiy et al. (2021).

For our experiments, we double the number of heads while fixing the width to ensure that any change in results is not due to the increased dimension of representations. In Tab. 6, it can be seen that even doubling the number of heads in ViTs only marginally helps in

Table 6: Role of attention heads: for the Split CIFAR-100 benchmark, increasing the number of attention heads (while fixing the total width), does not impact the performance significantly.

| Model | Heads | Params (M) | Average Accuracy | Average Forgetting | Learning Accuracy |
|---|---|---|---|---|---|
| ViT 512/1024 | 4 | 8.8 | 50.9 ±0.73 | 23.8 ±1.3 | 72.8 ±6.13 |
| ViT 512/1024 | 8 | 8.8 | 51.7 ±1.4 | 21.9 ±1.3 | 71.4 ±5.52 |
| ViT 1024/1536 | 4 | 30.7 | 57.4 ±1.59 | 14.4 ±1.96 | 66.0 ±5.89 |
| ViT 1024/1536 | 8 | 30.7 | 60.4 ±1.56 | 12.2 ±1.12 | 67.4 ±5.57 |

increasing the learning accuracy and lowering the forgetting. This suggests that increasing the number of attention heads may not be an efficient approach for improving continual learning performance in ViTs. We, however, note that consistent with other observations in the literature (Paul & Chen, 2022), ViTs show promising robustness against distributional shifts as evidenced by lower forgetting numbers, for the same amount of parameters, on the Split CIFAR-100 benchmark (*cf.* Fig. 1c).

**Conclusion.** While increasing the number of attention heads can improve the performance, the gain is not substantial. We believe increasing the hidden dimension of ViTs is a better approach for improving the performance if our computational budget allows for it.

## 4 PRACTICAL SUGGESTIONS FOR IMPROVING ARCHITECTURES

We now provide practical suggestions to improve the performance of architectures, that are derived from our results in Sec. 3. For CNNs, we add BatchNormalization and MaxPooling, as they both are shown to improve the learning ability of the model and skip connections that make the optimization problem easier. Tab. 7 shows the results on both CIFAR-100 and ImageNet-1K. It can be seen from the table that adding these components significantly improves the CNNs performance, as evidenced by improvement over almost all the metrics, including average accuracy. In addition, motivated by Sec. 3.1, we can use wider CNNs that have less forgetting and higher accuracy. For ResNets, we either remove the GAP, as is the case with CIFAR-100, or locally average the features in the penultimate layer by a 4x4 filter, as is the case with ImageNet-1K. The reason for not fully removing the average pooling from the ImageNet-1K is the resultant large increase in the number of parameters in the classifier layer. It can be seen from Tab. 7 that fully or partially removing the GAP, CIFAR-100 or ImageNet-1K, respectively, highly improves the retention capability of ResNets, indicated by their lower forgetting.

Table 7: Improving CNNs and ResNets architectures on both Split CIFAR-100 and ImageNet-1K.

| Model | Benchmark | Params (M) | Average Accuracy | Average Forgetting | Learning Accuracy |
|---|---|---|---|---|---|
| CNN x4 | CIFAR-100 | 2.3 | 68.1 ±0.5 | 8.7 ±0.21 | 76.4 ±6.92 |
| CNN x4 + BN + MaxPool + Skip | CIFAR-100 | 1.5 | 77.7 ±0.77 | 6.5 ±0.58 | 83.7 ±6.31 |
| CNN x8 | CIFAR-100 | 7.5 | 69.9 ±0.62 | 8.0 ±0.71 | 77.5 ±6.78 |
| CNN x8 + BN + MaxPool + Skip | CIFAR-100 | 6.1 | 78.1 ±1.15 | 5.7 ±0.36 | 83.3 ±6.27 |
| CNN x3 | ImageNet-1K | 9.1 | 63.3 ±0.68 | 5.8 ±0.93 | 71.6 ±2.31 |
| CNN x3 + BN + MaxPool + Skip | ImageNet-1K | 9.1 | 66.4 ±0.47 | 5.4 ±0.3 | 74.7 ±2.1 |
| CNN x6 | ImageNet-1K | 24.2 | 66.7 ±0.62 | 3.9 ±0.86 | 70.1 ±3.21 |
| CNN x6 + BN + MaxPool + Skip | ImageNet-1K | 24.3 | 72.1 ±0.41 | 4.0 ±0.22 | 75.7 ±2.57 |
| ResNet-18 | CIFAR-100 | 11.2 | 45.0 ±0.63 | 36.8 ±1.08 | 74.9 ±3.98 |
| ResNet-18 w/o GAP | CIFAR-100 | 11.9 | 67.4 ±0.76 | 11.2 ±1.98 | 74.2 ±4.79 |
| ResNet-50 | CIFAR-100 | 23.6 | 56.2 ±0.88 | 9.5 ±0.38 | 67.8 ±5.09 |
| ResNet-50 w/o GAP | CIFAR-100 | 26.7 | 71.4 ±0.29 | 6.6 ±0.12 | 73.0 ±5.18 |
| ResNet-34 | ImageNet-1K | 21.8 | 62.7 ±0.53 | 19.0 ±0.67 | 80.4 ±2.57 |
| ResNet-34 w 4x4 AvgPool | ImageNet-1K | 23.3 | 66.0 ±0.24 | 4.2 ±0.16 | 70.2 ±3.87 |
| ResNet-50 | ImageNet-1K | 25.5 | 66.1 ±0.69 | 17.3 ±0.58 | 83.3 ±1.57 |
| ResNet-50 w 4x4 AvgPool | ImageNet-1K | 31.7 | 67.2 ±0.13 | 3.5 ±0.35 | 72.8 ±3.27 |

## 5 RELATED WORK

**Algorithms.** Perhaps the most related side of algorithms to our work is the *parameter-isolation* methods where different parts of the model are devoted to a specific task (Yoon et al., 2018; Mallya & Lazebnik, 2018; Wortsman et al., 2020). Often, these algorithms start with a fixed-size model, and the algorithm aims to allocate a subset of the model for each task such that only a specific part of the model is responsible for the knowledge for that task. For instance, PackNet (Mallya & Lazebnik, 2018) uses iterative pruning to free space for future tasks, while (Wortsman et al., 2020) propose to fix with a randomly initialized and over-parameterized network to find a binary mask for each task. Nevertheless, even though the focus of these methods is the model and its parameters, the main objective is to have an algorithm that can use the model as efficiently as possible. Consequently, the significance of the architecture is often overlooked. We believe our work is orthogonal to the algorithmic side of continual learning research as any of the methods mentioned above can use a different architecture for their proposed algorithm, as we have shown in Fig. 1a and Tab. 8.

**Architectures.** There have been some efforts on applying architecture search to continual learning (Xu & Zhu, 2018; Huang et al., 2019; Gao et al., 2020; Li et al., 2019; Lee et al., 2021). However, when it comes to architecture search, the current focus of continual learning literature is mainly on deciding how to efficiently share or expand the model by delegating these decisions to the controller. For instance, Lee et al. (2021) use a ResNet as the base model and propose a search algorithm that decides whether the parameters should be reused or new parameters are needed for the new task, and the implications of architectural decisions have not been discussed. We believe future works that focus on the architecture search can benefit from our work by designing better search spaces that include various components that we have shown are important in continual learning. Finally, beyond architecture search methods, there are several works that focus on the non-algorithmic side of CL. Mirzadeh et al. (2022) study the impact of width and depth on forgetting and show that as width increases, the forgetting will decrease. Our work extends their analysis for larger-scale ImageNet-1K benchmark, more architectures, and more setups. Recently, Ramasesh et al. (2022) shows that in the pre-training setups, the scale of model and dataset can improve CL performance. In this work, however, we study the learning and retention capabilities of various architectures when models are trained from scratch. However, we show that while scaling the model can be helpful, the impact of architectural decisions is more significant. For instance, as shown in Tab. 1 and Tab. 2b, the best performing architectures are not the ones with highest number of parameters.

## 6 DISCUSSION

In this work, we have studied the role of architecture in continual learning. Through extensive experiments on a variety of benchmarks, we have shown that different neural network architectures have different learning and retention capabilities, and a slight change in the architecture can result in a significant change in performance in a continual learning setting. While our work has limitations regarding the studied domains, we believe our results show promising directions for future works and add novel insights over the existing works in this space.

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

## APPENDIX OVERVIEW

In this document, we provide additional details regarding the main work. More specifically:

- First, in Sec. A we cover the details regarding our experimental setup, design choices, architecture details, and hyper-parameters.
- Second, in Sec. B we provide additional experiments and results and a more detailed version of some of the experiments.

## A  EXPERIMENTAL SETUP DETAILS

### A.1  ARCHITECTURES

#### A.1.1  CNNs

Unless otherwise stated, the CNN models in this work solely include the convolutional layers, followed by a single feed-forward layer for classification. All CNNs use the ReLU activation function and use 3x3 kernels and a stride of 2.

For the CIFAR-100 experiments, the *base CNN* contains 3 convolutional layers with 32, 64, and 128 channels respectively, and *CNN* $\times N$ refers to the base CNN model where the number of channels in each layer is multiplied by $N$. Hence, *CNN* $\times 4$ refers to 3 convolutional layers with 128, 256, and 512 channels, followed by a feed-forward classification layer. In Split ImageNet-1K experiments, the *base CNN* contains 6 convolutional layers with 64 channels for the first four layers and 128 channels for the last two layers. Similar to the setup for the Split CIFAR-100 experiments, all models have a single-layer feed-forward layer for classification and use a kernel size of 3, with a stride of 2.

When we use pooling layers in CNNs(e.g., Sec. 3.4), to keep the dimension of features the same with the CNNs that don't have pooling layers, we use a stride of 1 for convolutional layers. In other words, every convolutional layer leads to a reduced feature map by a factor of 2: either the convolution has stride 2 (e.g., CNNs in Tab. 1) or else it has a stride 1, followed by a pooling layer (e.g., CNN+(Avg/Max)Pool in Tab. 4 and Tab. 7). Finally, for the experiments with skip-connections added, we add 2 skip-connections (with projections) for the 3-layer CNNs on Split CIFAR100 that adds shortcuts from layer 1 to output of layer 2 and layer 2 to the output of layer 3.

#### A.1.2  RESNETS

The ResNets we use in the ImageNet experiment are the standard models, and we do not modify them. However, the ResNets in the CIFAR-100 experiments use $3 \times 3$ kernels with a stride of 1, rather than the $7 \times 7$ kernels with a stride of 2. This is a common practice for the ResNet models for low-dimensional CIFAR images since it does not reduce the dimension of input significantly. Other than this modification for CIFAR-100 experiments, the rest of the ResNet architecture kept the same.

#### A.1.3  WIDERESNETS

The WideResNets (WRN) models on both CIFAR-100 and ImageNet benchmarks follow the original implementation of WRN models. For all experiments, we use a dropout factor of 0.1 as we empirically observed increasing the dropout does not improve the performance significantly.

#### A.1.4  VISION TRANSFORMERS

The Vision Transformer (ViT) models in our ImageNet experiment follow the same architecture as the original vision transformers. Similar to the original ViT models, we use the patch size of 16 for both ViT models in our ImageNet-1K experiments.

However, since the original vision transformer paper does not provide the details for the best practices for the CIFAR benchmark, we used smaller versions of the ViTs to match the training cost of other models. In those experiments, *ViT 512-1024* refers to a ViT model with 4 layers, with a width of 512 and MLP size of 1024. Similarly, *ViT 1024-1536* has a width of 1024 with the MLP hidden size of 1536. All models use the patch size of 4 (i.e., 64 patches for CIFAR images), but we empirically observed increasing the patch size to 8 does not impact the results significantly.

## A.2 HYPERPARAMETERS

In this section, we report the hyper-parameters we used for our experiments. We include the chosen hyper-parameter for each architecture in parentheses.

### A.2.1 ROTATED MNIST

We follow the setting in (Mirzadeh et al., 2022) for our MNIST experiments in Sec. 3.1.

```
learning rate: [0.001, 0.01 (MLP), 0.05, 0.1]
momentum: [0.0 (MLP), 0.8]
weight decay: [0.0 (MLP), 0.0001]
batch size: [16, 32, 64 (MLP)]
```

### A.2.2 SPLIT CIFAR-100

We use the following grid of hyper-parameters for the CIFAR-100 experiments:

```
learning rate: [0.001, 0.005, 0.01 (ViT 1024, ResNet 50/101), 0.05 (CNNs, ResNet 18/34, ViT 512, WRNs), 0.1]
learning rate decay: [1.0 (ResNet 50/101), 0.8 (CNN, ViTs, ResNet 18/34, WRNs)]
momentum: [0.0 (CNN, ViTs, ResNet 50/101), 0.8 (ResNet 18/34, WRNs)]
weight decay: [0.0 (CNNs, ViTs), 0.0001 (ResNets, WRNs)]
batch size: [8 (CNNs, ResNet18/34), 16 (WRNs), 64 (ResNet 50/101, ViT 512), 128 (ViT 1024)]
```

We note that the learning rate decay is applied after each task.

### A.2.3 SPLIT IMAGENET-1K

Due to computation budget, we use smaller a smaller grid for the ImageNet-1K experiments. However, we make sure that the grid is diverse enough to cover various family of architectures.

```
learning rate: [0.005, 0.01, 0.05, 0.1 (All models)]
learning rate decay: [1.0, 0.75 (All models)]
momentum: [0.0, 0.8 (All models)]
weight decay: [0.0 (CNNs), 0.0001 (ResNets, WRN, ViTs)]
batch size: [64 (All models), 256]
```

## A.3 TRAINING INFRASTRUCTURE

For each run of the MNIST and CIFAR-100 experiments, we have used 1 NVIDIA V-100 GPU. For the ImageNet-1K experiments, we have used 32 TPU v2 or v3. Moreover, we have used JaxBradbury et al. (2018), Haiku Hennigan et al. (2020), and Optax Hessel et al. (2020) for the implementation.

# B ADDITIONAL RESULTS

## B.1 ALGORITHMS VS. ARCHITECTURES

In Fig. 1a, we have provided the results for the ResNet-18 model. Here, we provide the average accuracy and average forgetting for various models and algorithms on split CIFAR-100 with 20 tasks.

Table 8: Different Algorithms and Architectures

| Algorithm | Parameters (M) | Architecture | Average Accuracy | Average Forgetting |
|---|---|---|---|---|
| Finetune | 11.2 | ResNet-18 | 45.0 ±0.63 | 36.8 ±1.08 |
| EWC | 11.2 | ResNet-18 | 50.6 ±0.70 | 26.6 ±2.53 |
| AGEM (Mem = 1000) | 11.2 | ResNet-18 | 61.8 ±0.45 | 22.9 ±1.59 |
| ER (Mem = 1000) | 11.2 | ResNet-18 | 64.3 ±0.99 | 19.7 ±1.26 |
| Finetune | 11.9 | ResNet-18 (w/o GAP) | 67.4 ±0.76 | 11.2 ±1.98 |
| AGEM (Mem = 1000) | 11.9 | ResNet-18 (w/o GAP) | 72.8 ±1.33 | 5.8 ±0.39 |
| ER (Mem = 1000) | 11.9 | ResNet-18 (w/o GAP) | 74.4 ±0.33 | 4.6 ±0.54 |
| Finetune | 2.3 | CNN x4 | 68.1 ±0.50 | 8.7 ±0.21 |
| ER (Mem = 1000) | 2.3 | CNN x4 | 74.4 ±0.27 | 2.4 ±0.12 |

We remind the reader that the aim of our work is not to undermine the importance of continual learning algorithms. On the contrary, we appreciate the recent algorithmic improvements in the continual learning literature. The aim of this work is to show that *the role of architecture is significant, and a good architecture can complement a good algorithm in continual learning*.

## B.2    BATCHNORM STATISTICS ON SPLIT CIFAR-100

We show the first layer's BN statistics for CNN×4 in Fig. 3 using kernel density estimation with Gaussian kernel. As illustrated, the batch statistics and learned BN parameters do not change significantly across different tasks. Here, for simplicity, we have shown only four tasks from the beginning, middle, and late of the learning experience. Moreover, we focus on the first layer's statistics, which is the first layer of the model that operates on the data.

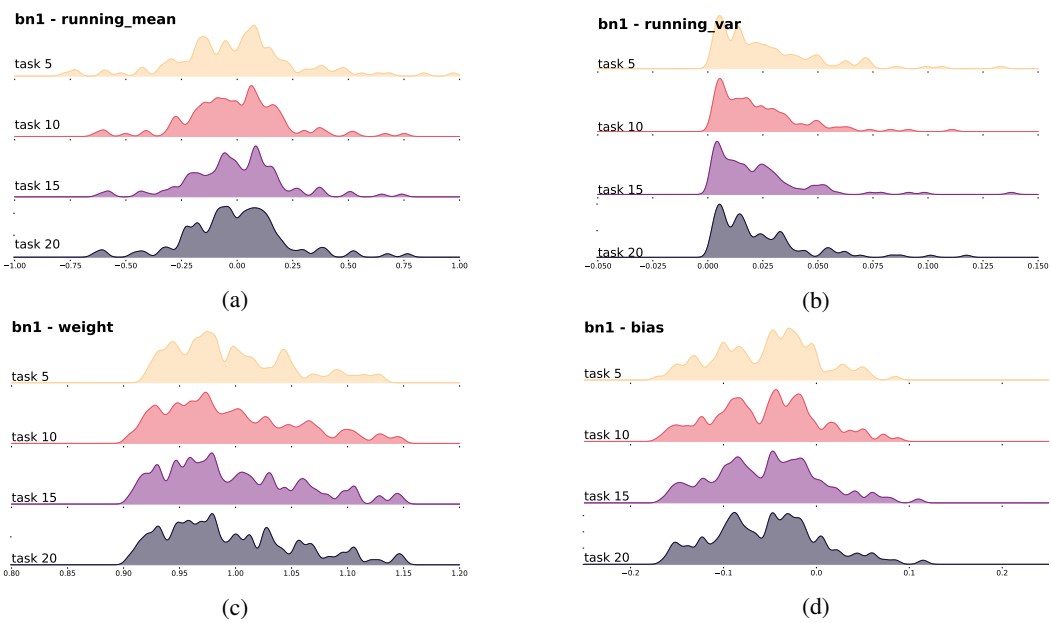

Figure 3: BN statistics for the first layer of CNN×4 on Split CIFAR-100: the statistics do not change significantly throughout the continual learning experience.

## B.3    BATCHNORM STATISTICS ON PERMUTED MNIST

In Sec. 3.2, we have discussed that if the batch statistics change significantly across tasks, batch normalization can increase the forgetting. To this end, we design an experiment where we train two MLP-128 networks (with and without BN) on the Permuted MNIST benchmark(Goodfellow et al., 2014) with five tasks. While Permuted MNIST is not a very realistic benchmark, it fits our requirements for synthetic distribution shift (i.e., shuffling pixels).

While Tab. 9 demonstrates the benefit of adding BN (i.e., improving learning accuracy), we can observe a significant increase in average forgetting as well. To investigate this more, we visualize the BN statistics in Fig. 4 where we can see compared to Fig. 3, the statistics change more, confirming our hypothesis in Sec. 3.2.

Table 9: Permuted MNIST: The MLP with BN has slightly higher learning accuracy, but significantly higher forgetting as well.

| Model | Average Accuracy | Average Forgetting | Learning Accuracy |
|---|---|---|---|
| MLP-128 | 86.8 ±0.95 | 10.9 ±0.88 | 95.5 ±0.33 |
| MLP-128 + BN | 73.2 ±0.82 | 32.5 ±0.72 | 97.8 ±0.45 |

## B.4    MORE NUMBER OF EPOCHS ON SPLIT CIFAR-100

While in our main text, we have used 10 epochs for Split CIFAR-100, here in Tab. 10 we show that the main conclusions hold even when we train the models longer. More specifically, we can see that wider models perform better in terms of learning and retention capabilities, batch normalization and max pooling improve the learning ability of the models, and removing the global average pooling can improve the performance of ResNets.

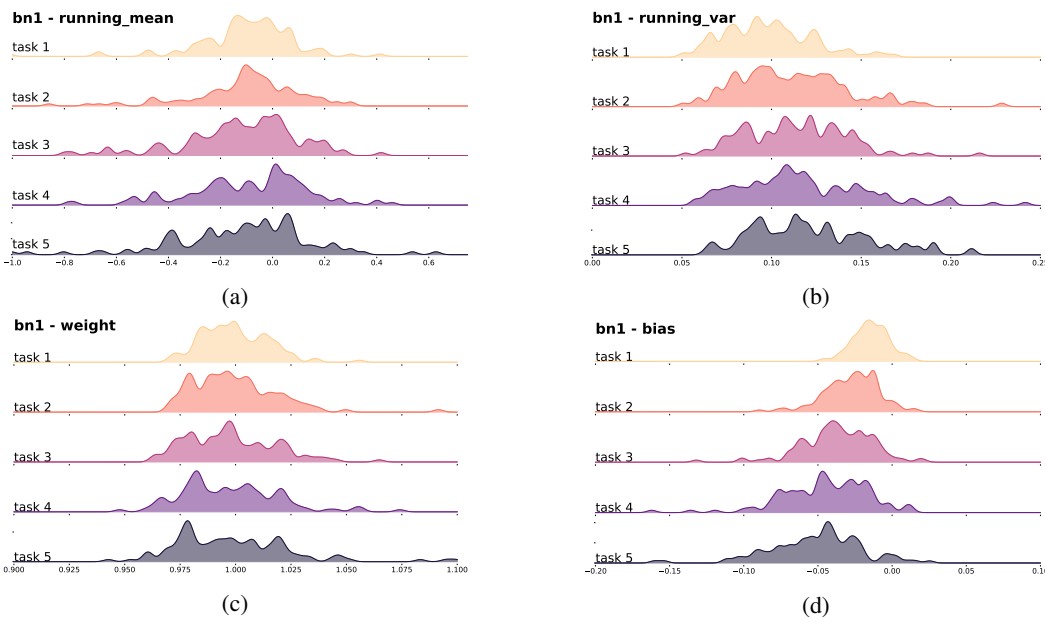

Figure 4: BN statistics for the first layer of MLP-128 on Permuted MNIST: the statistics change more compared to Fig. 3

Interestingly, we can see that simple MLPs (with two hidden layers), perform surprisingly well and demonstrate a comparable performance in terms of forgetting with CNNs. However, as it is well known in the single task setup, they are not great learners of image data, and their learning accuracy is often much smaller than their convolutional models. But, it is worth mentioning that these simple MLPs still outperform the commonly used ResNets (with GAP layer) in this task-incremental benchmark, which is another strong evidence of the significance of architecture in continual learning.

Table 10: Comparing the impact of components in with two different settings: The components that are helpful in the short training time (e.g., removing GAP layers, adding pooling or batch norm layers), are also beneficial when the training time is longer.

| Model | Params (M) | Epochs = 10 | | | Epochs = 50 | | |
|---|---|---|---|---|---|---|---|
| | | Average Accuracy | Average Forgetting | Learning Accuracy | Average Accuracy | Average Forgetting | Learning Accuracy |
| ResNet-18 | 11.2 | $45.0 \pm 0.63$ | $36.8 \pm 1.08$ | $74.9 \pm 3.98$ | $37.1 \pm 0.59$ | $48.9 \pm 1.35$ | $82.4 \pm 4.83$ |
| ResNet-18 w/o GAP | 11.9 | $67.4 \pm 0.76$ | $11.2 \pm 1.98$ | $74.2 \pm 4.79$ | $66.1 \pm 0.44$ | $17.3 \pm 1.43$ | $80.2 \pm 4.77$ |
| ResNet-50 | 23.6 | $56.2 \pm 0.88$ | $9.5 \pm 0.38$ | $67.8 \pm 5.09$ | $53.4 \pm 0.29$ | $14.5 \pm 0.64$ | $75.3 \pm 5.65$ |
| ResNet-50 w/o GAP | 26.7 | $71.4 \pm 0.29$ | $6.6 \pm 0.12$ | $73.0 \pm 5.18$ | $71.2 \pm 0.18$ | $7.3 \pm 0.22$ | $76.5 \pm 4.87$ |
| MLP-512 | 1.9 | $53.9 \pm 0.99$ | $14.0 \pm 1.41$ | $63.2 \pm 6.35$ | $53.7 \pm 1.06$ | $17.2 \pm 1.51$ | $64.7 \pm 5.77$ |
| MLP-512 + BN | 1.9 | $58.2 \pm 0.55$ | $12.8 \pm 0.71$ | $69.5 \pm 4.11$ | $57.0 \pm 0.43$ | $17.6 \pm 0.47$ | $72.0 \pm 4.25$ |
| MLP-1024 | 4.3 | $57.1 \pm 0.4$ | $9.0 \pm 1.11$ | $65.3 \pm 2.42$ | $57.5 \pm 1.18$ | $13.6 \pm 1.36$ | $73.4 \pm 3.41$ |
| MLP-1024 + BN | 4.3 | $62.0 \pm 0.66$ | $8.3 \pm 0.74$ | $71.1 \pm 2.48$ | $61.5 \pm 0.5$ | $13.5 \pm 1.37$ | $72.8 \pm 2.68$ |
| CNN x4 | 2.3 | $68.1 \pm 0.5$ | $8.7 \pm 0.21$ | $76.4 \pm 6.92$ | $62.6 \pm 0.4$ | $14.4 \pm 0.62$ | $75.2 \pm 6.25$ |
| CNN x4 + BN | 2.3 | $74.0 \pm 0.56$ | $8.1 \pm 0.35$ | $81.7 \pm 6.68$ | $68.9 \pm 0.93$ | $13.8 \pm 0.68$ | $80.7 \pm 5.83$ |
| CNN x4 + Maxpool | 2.3 | $74.4 \pm 0.34$ | $9.3 \pm 0.47$ | $83.3 \pm 6.1$ | $69.3 \pm 0.79$ | $13.5 \pm 0.85$ | $81.9 \pm 5.47$ |
| CNN x8 | 7.5 | $69.9 \pm 0.62$ | $8.0 \pm 0.71$ | $77.5 \pm 6.78$ | $64.3 \pm 0.82$ | $13.2 \pm 1.01$ | $78.8 \pm 6.61$ |
| CNN x8 + BN | 7.5 | $76.1 \pm 0.3$ | $5.9 \pm 0.16$ | $81.7 \pm 6.83$ | $71.7 \pm 0.79$ | $11.5 \pm 0.85$ | $82.4 \pm 6.18$ |
| CNN x8 + Maxpool | 7.5 | $77.2 \pm 0.53$ | $7.1 \pm 0.33$ | $84.0 \pm 5.81$ | $73.6 \pm 2.25$ | $12.9 \pm 1.07$ | $84.4 \pm 5.06$ |

### B.5 PRELIMINARY RESULTS ON CLASS-INCREMENTAL LEARNING

While exploring all scenarios with all architectures is beyond the scope of this work, as a proof of concept, in Tab. 11, we report the results (over five runs) on Split CIFAR-10 with 5 tasks (5 epochs per task) in the class-incremental scenario . Most importantly, we can observe:

1. The architectures (e.g., CNN vs ResNet) play a significant role as we can see by various metrics.

2. Architectural decisions (e.g., removing GAP, adding MaxPooling) are beneficial and they can significantly increase the learning and/or retention capabilities of models.

Table 11: The class-incremental scenario on 5-split CIFAR-10: Similar to the task-incremental scenario, we can observe the significant impact of architectural changes on continual learning performance.

| Model | Params (M) | Average Accuracy | Average Forgetting | Learning Accuracy |
|---|---|---|---|---|
| CNNx4 | 1.6 | 62.0±(1.41) | 15.6±(0.25) | 74.4±(1.61) |
| CNNx8 | 6.2 | 66.5±(1.23) | 12.1±(0.5) | 75.2±(0.83) |
| CNNx4 + MaxPool | 1.6 | 69.1±(0.45) | 15.9±(1.07) | 82.0±(1.09) |
| CNNx8 + MaxPool | 6.2 | 72.5±(0.18) | 12.8±(0.91) | 84.3±(0.91) |
| ResNet-18 | 11.1 | 62.1±(0.2) | 37.7±(1.08) | 92.2±(0.68) |
| ResNet-18 w/o GAP | 11.2 | 67.6±(0.84) | 17.5±(0.92) | 81.5±(1.57) |
| ViT 512/1024 | 8.5 | 64.3±(0.1) | 20.5±(1.07) | 83.1±(0.8) |

Overall, while our small-scale experiment is limited to a specific setup, still, the conclusions are in line with our results on task-incremental setup.

