# OpenReview forum: "Architecture Matters in Continual Learning"
_ICLR.cc/2023/Conference — Submitted to ICLR 2023_

### Official Review · Reviewer_b2F4 · 2022-10-24

**Confidence:** 5
**Correctness:** 2
**Technical Novelty And Significance:** 2
**Empirical Novelty And Significance:** 3
**Recommendation:** 3

**Clarity, Quality, Novelty And Reproducibility:**

- Very good clarity of writing.
- Experiments are mostly well-run in what they do (with hyperparameters reported). But some baselines and comparisons are missing that I view as crucial (Joint accuracy baselines for all tables, as well as significantly more results with a continual learning algorithm that is not fine-tuning).
- I am not aware of papers specifically looking at architecture choices such as BatchNorm and Skip connections for continual learning. Depth and width of network has been considered before at a smaller scale.

**Strength And Weaknesses:**

**Strengths**

1. Many experiments are run, including on ImageNet-1k, which is large-scale for continual learning.

2. The paper is well-written: it reads well and the paper structure is good.

3. I am not aware of many other papers that look at how changing the architecture impacts continual learning performance (aside from the papers cited in the Introduction of this paper).

4. I liked the style and content of the 'Limitations' paragraph in the Introduction.

5. I like the result of Figure 1a: ResNet-18 without GAP layer but with fine-tuning outperforms EWC (and ER with 1000 samples) on standard ResNet-18.


**Weaknesses (especially first 3 points)**

6. This paper (almost) exclusively uses fine-tuning as the 'continual learning' algorithm to test different architectures. This is a problem because I am not sure if results with fine-tuning translate to results with other algorithms like EWC, ER, or more recent algorithms. As a random representative example: perhaps, using EWC with skip connections improves performance considerably over EWC without skip connections? Other algorithms are considered in Figure 1a and Appendix B1, but I believe such results need to be throughout the entire paper. In Appendix B1, the authors say that 'a good architecture can complement a good algorithm in continual learning': please convince me that this is indeed the case!

7. The authors draw a lot of conclusions from changing an architecture choice and looking at continual learning metrics. For example, with skip connections and pooling layers, they say that there is not much improvement in continual learning. However, I find this misleading as the joint accuracy on this benchmark also does not improve significantly. Previous works (cited in the paper) argue that these architecture changes are important for image classification / joint accuracy in general, but this is clearly not the case with the benchmark used. So I cannot tell if these architecture changes are helpful or not in continual learning: ideally, they would help in joint accuracy before drawing conclusions about continual learning. For MaxPool, the joint accuracy increases by a similar amount as the average accuracy. (I note that the authors do not claim anything specific in the text, but I view these results as not very helpful/useful when it comes to architecture design for continual learning.)

8. Following up from this point, the 'joint accuracy' baseline is missing in many of the tables/results. I want to *always* see if joint accuracy increases with an architecture change, and how this change correlates with the continual learning average accuracy change. Else, any benefits in changing the architecture could just be due to the joint accuracy increasing, which says the importance of this architecture change to the specific datasets, but not to the ability of a network to perform a good stability-plasticity trade-off in general.

9. Section 3.1 is about width and depth of neural networks, which was already studied in Mirzadeh et al. There is no new insight or takeaway in this paper. I agree that this paper's results verify the same conclusions on more benchmarks and architectures, however, this is not too significant by itself.

10. Removing GAP in Table 5 improves results, but also drastically increased number of parameters. This is an interesting result but can the authors dig deeper? For example, CNNx8 has worse average accuracy than CNNx4 (16x) + GAP, despite having twice the pre-classification width. The results do not seem conclusive enough to me that GAP only reduces performance because it reduces the last-layer width.

**Summary Of The Paper:**

This paper empirically looks at how changing the neural network architecture can impact continual learning performance. For example, if changing width/depth of the network, or adding BatchNorm layers, or adding skip connections, changes how well the network can learn continually. Many experiments are performed on Split CIFAR-100 and Split ImageNet-1k, which is a large-scale benchmark. Recommendations are made based off empirical results.

**Summary Of The Review:**

Unfortunately points 6, 7, and 8 in weaknesses are very important for me and therefore I recommend reject. I am not convinced that these results will hold for continual learning algorithms that are not fine-tuning. I also need to see and compare all results with Joint accuracy. Therefore I do not agree with many of the claims made in the abstract.

---

### Official Review · Reviewer_4mDG · 2022-10-25

**Confidence:** 4
**Correctness:** 4
**Technical Novelty And Significance:** 2
**Empirical Novelty And Significance:** 4
**Recommendation:** 8

**Clarity, Quality, Novelty And Reproducibility:**

The clarity and quality of the paper is exceptionally good. The authors were able to clearly state the goals and evidence to support claims accurately. The structure of the paper is well done and easy to follow.

Novelty wise, there is no particular new method that is proposed. The main novelty lies on the new analysis/insights that are given by the paper. To the best of my knowledge, such good analysis is new. That is, the empirical novelty of this paper is high.

Reproducibility: the paper and appendix gives enough details of the implementation, hyper-parameters. I think it should be not difficult to reproduce the results.

**Strength And Weaknesses:**

Strength:

1. Well-organized goals and supportive experiments. It is a pleasure to read this paper because all of the factors that the author wish to study are well organized, with direct experiments to support the conclusions. Experiments are particularly supportive because 1) the metrics chosen (average forgetting, average accuracy, learning accuracy) are meaningful for CL; 2) direct controlled experiments to study each individual factor.

2. Great pratical suggestions summarized for researchers to be able to use the conclusions in this paper. For example, from the paper, it is clear that "ResNets and WRNs have better learning abilities, whereas CNNs and ViTs have better retention abilities. Simple CNNs achieve the best trade-off between learning and retention."

3. This architecture study can complement algorithm study. From Appendix B.1. It shows briefly that CL algorithms can further improve from the arhictecture that is chosen by this paper.



Weakness:
Not exactly weakness, but I am curious of the conclusions with other more recent architectures, and other tasks. The authors do not need to provide these in the rebuttal. If the evaluation pipeline code can be released publicly, it would be even more impactful (other researchers can evaluate those new architectures and contribute).
Another thing is that how CL algorithm and architecture impact each other is also an interesting future work, from a pratical view.

**Summary Of The Paper:**

This paper studies the architecture choices in the context of countinual learning (CL). It

1) compares different popular architectures
2) study component choices for these architectures, such as width/depth, BN, skip, pooling etc.
3) draw conclusions from the study to make pratical suggestions.

The experiments are on Rotated MNIST, CIFAR-100 Split and ImageNet-1k Split.


**Summary Of The Review:**

Overall the paper is well-written. The studies done is novel and very informative. In particular, authors draw conclusions that have good empirical evidence and also point out things that do not have enough evidence to support. The suggestions given are also pratically useful for other researchers. I recommend accept.

---

### Official Review · Reviewer_y8Pr · 2022-10-25

**Confidence:** 5
**Correctness:** 3
**Technical Novelty And Significance:** 2
**Empirical Novelty And Significance:** 2
**Recommendation:** 5

**Clarity, Quality, Novelty And Reproducibility:**

Clarity and quality of the work is good but more work needs to be done in regards to novelty and reproducibility

**Strength And Weaknesses:**

Strengths:

•	Architecture choice is a key component of the modeling process and can have a considerable effect on the continual learning performance. This is an important area of research that needs to be considered.

•	The experiments within the considered architectures have been done thoroughly with multiple random initializations, and corresponding best hyperparameters. The datasets are also well suited for the analysis did in this work.

•	It is interesting to see the effect of various architecture components such as width, depth, batchnorm, skip connections, pooling layers and attention heads.

Limitations:

•	Only Task-incremental has been primarily studied and not the class-incremental learning setting which is a harder problem

•	The effect of width and depth have already been studied in Mirzadeh et al. 2022, and effect of normalizing schemes has been
studied in Pham et al 2022, effect of distribution shift is partially studied in Paul and Chen 2022, with these having similar conclusions as this work.

•	Effect of skip connections is not fully developed. For example, skip connections can be applied at different spans of layers. It is not clear what is used in this work.

•	The explanation of why max pooling helps is speculative, visualizing the activations might be helpful instead of speculating that low-level feature might have been effectively learned and helped generalization.

•	Majority of the work looks at naïve training in the continual learning setting but the effect of the architecture choice on various classes of continual learning is not fully developed (table 8). I was expecting to see how for example the architecture choices effect the experience-replay methods etc., which will be more useful for future algorithm development. The approaches considered in table 8 are also not the state of the art. If the state-of-the-art approaches are not sensitive to architecture choice then this work will not be useful.

•	Not clear how the hyperparameter search is done differently for continual learning compared to the single-task learning.


**Summary Of The Paper:**

This work looks at empirical analysis of the effect of architecture choice and components on continual learning performance and draws a few insights on best practices for architecture selection.

**Summary Of The Review:**

Effect of architecture choice on continual learning is an important area of research. A more thorough analysis needs to be done especially connecting the architecture sensitivity to the classes of continual learning approaches to make this work actionable and improve the novelty aspect which currently is less due to similarity with other sensitivity analysis works

---

### Decision · Program_Chairs · 2023-01-20

**Decision:**

Reject

**Justification For Why Not Higher Score:**

The paper is simply inconclusive and incomplete at the current stage, as it does not reveal much about how architecture impacts different continual learning algorithms.

**Justification For Why Not Lower Score:**

N/A

**Metareview: Summary, Strengths And Weaknesses:**

The paper studies the impact of the choice of the architecture on its continual learning performance, by examining average accuracy, learning accuracy, and forgetting of diverse architectures (e.g. ViT, WideResNet with different width and depths) with different architectural components (e.g. BN, skip-connections, GAP layer) on multiple benchmark datasets. From the experimental results, the authors identify the architectures with the highest learning accuracy, or lowest forgetting, which may be useful for practitioners. Another contribution is the experimental validation on a new benchmark setting with a large dataset, ImageNet 1K Split.

The paper received split reviews (two negative and one positive). The reviewers in general found the tackled problem to be relevant for research on continual learning, the experimental results to be thorough, useful for practice, and paper well-written. However, two of the reviewers had serious concerns regarding the contribution and the novelty of the work over existing works such as [Mirzadeh et al. 22] and [Paul and Chen 22] which present similar findings, inconclusive impact of certain architectural components, and the lack of experiments with various continual learning algorithms, such as EWC, ER, and many others. Among these issues, I believe that the lack of experiments with continual learning algorithms (although partial results are provided in Table 8 of the appendix), is the most serious issue. The empirical study of different algorithms with different architectures is what the readers expect when they first see the title, and it is what can give the researchers and practitioners useful insights on how to overcome the sensitivity of existing algorithms to architectures, or which algorithms to choose, for practical continual learning problems. Since no one will perform continual finetuning in practical continual learning scenarios, the current set of results are not particularly useful.

I understand that the authors wanted to further study this aspect as they provide results in Table 8 of the appendix but maybe could not complete the experiments due to limited time, and suggest them to continue with their exploration of various continual learning methods before resubmitting the work to another venue.

[Mirzadeh et al. 22] Wide Residual Networks Forget Less Catastrophically, ICLR 2022
[Paul and Chen 22] Vision Transformers are Robust Learners, AAAI 2022